# Learning to Screen

**Alon Cohen**[*]    **Avinatan Hassidim**[†]    **Haim Kaplan**[‡]    **Yishay Mansour**[§]    **Shay Moran**[¶]

## Abstract

Imagine a large firm with multiple departments that plans a large recruitment. Candidates arrive one-by-one, and for each candidate the firm decides, based on her data (CV, skills, experience, etc), whether to summon her for an interview. The firm wants to recruit the best candidates while minimizing the number of interviews. We model such scenarios as an assignment problem between items (candidates) and categories (departments): the items arrive one-by-one in an online manner, and upon processing each item the algorithm decides, based on its value and the categories it can be matched with, whether to retain or discard it (this decision is irrevocable). The goal is to retain as few items as possible while guaranteeing that the set of retained items contains an optimal matching.

We consider two variants of this problem: (i) in the first variant it is assumed that the $n$ items are drawn independently from an unknown distribution $D$. (ii) In the second variant it is assumed that before the process starts, the algorithm has an access to a training set of $n$ items drawn independently from the same unknown distribution (e.g. data of candidates from previous recruitment seasons). We give near-optimal bounds on the best-possible number of retained items in each of these variants. These results demonstrate that one can retain exponentially less items in the second variant (with the training set).

Our algorithms and analysis utilize ideas and techniques from statistical learning theory and from discrete algorithms.

## 1 Introduction

Matching is the bread-and-butter of many real-life problems from the fields of computer science, operations research, game theory, and economics. Some examples include job scheduling where we assign jobs to machines, economic markets where we allocate products to buyers, online advertising where we assign advertisers to ad slots, assigning medical interns to hospitals, and many more.

One particular example that motivates this work is the following example from labor markets. Imagine a firm that is planning a large recruitment. Candidates arrive one-by-one and the HR department immediately decides whether to summon them for an interview. Moreover, the firm has multiple departments, each requiring different skills and having a different target number of hires. Different employees have different subsets of the required skills, and thus fit only certain departments and with a certain quality. The firm's HR department, following the interviews, decides which candidates to recruit and to which departments to assign them. The HR department has to maximize the total quality of the hired employees such that each department gets its required number of hires with the required skills. In addition, the HR uses data from the previous recruitment season in order to minimize the number of interviews while not compromising the quality of the solution.

---

[*]Technion—Israel Inst. of Technology and Google Research. `aloncohen@technion.ac.il`

[†]Bar-Ilan University and Google Research. `avinatanh@gmail.com`

[‡]Tel-Aviv University and Google Research. `haimk@tau.ac.il`

[§]Tel-Aviv University and Google Research. `mansour.yishay@gmail.com`

[¶]Princeton University. `shaymoran1@gmail.com`. This work was done while the author was working at Google Research.

We study the following formulation of the problem above. We receive $n$ items (candidates), where each item has a subset of $d$ properties (departments) denoted by $P_1, \ldots, P_d$. We select $k$ items out of the $n$, subject to $d$ constraints of the form

*exactly $k_i$ of the selected items must satisfy a property $P_i$,*

where $\sum_{i=1}^{d} k_i = k$ and we assume that $d \ll k \ll n$. Furthermore, if item $c$ possesses property $P_i$, then it has a value $v_i(c)$ associated with this property. Our goal is to compute a matching of maximum value that associates $k$ items to the $d$ properties subject to the constraints above.

We consider matching algorithms in the following online setting. The algorithms receive $n$ items online, drawn independently from $D$, and either reject or retain each item. Then, the algorithm utilizes the retained items and outputs an (approximately-)optimal feasible solution. We present a naive greedy algorithm that returns the optimal solution with probability at least $1 - \delta$ and retains $O(k \log(k/\delta))$ items in expectation. We prove that no other algorithm with the same guarantee can retain less items in expectation.

Thus, to further reduce the number of retained items, we add an initial preprocessing phase in which the algorithm learns an online policy from a *training set*. The training set is a single problem instance that consists of $n$ items drawn independently from the same unknown distribution $D$. We address the statistical aspects of this problem and develop efficient learning algorithms. In particular, we define a class of *thresholds-policies*. Each thresholds-policy is a simple rule for deciding whether to retain an item. We present uniform convergence rates for both the number of items retained by a thresholds policy and the value of the resulting solution. We show that these quantities deviate from their expected value by order of $\sqrt{k}$ (rather than an easier $\sqrt{n}$ bound; recall that we assume $k \ll n$) which we prove using concentration inequalities and tools from VC-theory. Using these concentration inequalities, we analyze an efficient online algorithm that returns the optimal offline solution with probability at least $1 - \delta$, and retains a near-optimal $O(k \log \log(1/\delta))$ number of items in expectation (compare with the $O(k \log(k/\delta))$ number of retained items when no training set is given).

**Related work.**   Our model is related to the online secretary problem in which one needs to select the best secretary in an online manner (see Ferguson, 1989). Our setting differs from this classical model due to the two-stage process and the complex feasibility constraints. Nonetheless, we remark that there are few works on the secretary model that allow delayed selection (see Vardi, 2015, Ezra et al., 2018) as well as matroid constraints [Babaioff et al., 2007]. These works differ from ours in the way the decision is made, the feasibility constraints and the learning aspect of receiving a single problem instance as a training example. Correa et al., 2018 consider a distributional setting for the *single-choice prophet inequality* problem. Similarly to the setting considered here, they assume that the data is drawn independently from an unknown distribution and that the algorithm has an access to a training-set sampled from the same distribution. However, the objective is quite different from ours: the goal is to pick a stopping time $\tau$ such that the $\tau$'th sample approximately maximizes the value among all samples (including those that were not seen yet).

Another related line of work in algorithmic economics studies the statistical learnability of pricing schemes (see e.g., Morgenstern and Roughgarden, 2015, 2016, Hsu et al., 2016, Balcan et al., 2018). The main difference of these works from ours is that our training set consists of a single "example" (namely the set of items that are used for training), and in their setting (as well as in most typical statistical learning settings) the training set consists of many i.i.d examples. This difference also affects the technical tools used for obtaining generalization bounds. For example, some of our bounds exploit Talagrand's concentration inequality rather than the more standard Chernoff/McDiarmid/Bernstein inequalities. We note that Talagrand's inequality and other advanced inequalities were applied in machine learning in the context of learning combinatorial functions [Vondrák, 2010, Blum et al., 2017]. See also the survey by Bousquet et al. [2004] or the book by Boucheron et al. [2013] for a more thorough review of concentration inequalities.

Furthermore, there is a large body of work on online matching in which the vertices arrive in various models (see Mehta et al., 2013, Gupta and Molinaro, 2016). We differ from this line of research, by allowing a two-stage algorithm, and requiring to output the optimal matching is the second stage.

Celis et al. [2017, 2018] studies similar problems of ranking and voting with fairness constraints. In fact, the optimization problem that they consider allows more general constraints and the value of a candidate is determined from votes/comparisons. The main difference with our framework is that

they do not consider a statistical setting (i.e. there is no distribution over the items and no training set for preprocessing) and focus mostly on approximation algorithms for the optimization problem.

## 2 Model and Results

Let $X$ be a domain of items, where each item $c \in X$ can possess any subset of $d$ properties denoted by $P_1, \ldots, P_d$ (we view $P_i \subseteq X$ as the set of items having property $P_i$). Each item $c$ has a value $v_i(c) \in [0, 1]$ associated with each property $P_i$ such that $c \in P_i$.

We are given a set $C \subseteq X$ of $n$ items as well as counts $k_1, \ldots k_d$ such that $\sum_{i=1}^{d} k_i = k$. Our goal is to select exactly $k$ items in total, constrained on selecting exactly $k_i$ items with property $P_i$. We assume that these constraints are *exclusive*, in the sense that each item in $C$ can be used to satisfy at most one of the constraints. Formally, a feasible solution is a subset $S \subseteq C$, such that $|S| = k$ and there is partition $S$ into $d$ disjoint subsets $S_1, \ldots, S_d$, such that $S_i \subseteq P_i$ and $|S_i| = k_i$. We aim to compute a feasible subset $S$ that maximizes $\sum_{i=1}^{d} \sum_{c \in S_i} v_i(c)$.

Furthermore, we assume that $d \ll k \ll n$. Namely, the number of constraints is much smaller than the number of items that we have to select, which is much smaller than the total number of items in $C$. In order to avoid feasibility issues we assume that there is a set $C_{\text{dummy}}$ that contains $k$ dummy 0-value items with all the $d$ properties (we assume that the algorithm has always access to $C_{\text{dummy}}$ and do not view them as part of $C$).

**Formulation as bipartite matching.** We first discuss the offline versions of these allocation problems. That is, we assume that $C$ and the capacities $k_i$ are all given as an input before the algorithm starts. We are interested in an algorithm for computing an optimal set $S$. That is a set of items of maximum total value that satisfy the constraints. This problem is equivalent to a maximum matching problem in a bipartite graph $(L, R, E, w)$ defined as follows.

- $L$ is the set of vertices in one side of the bipartite graph. It contains k vertices, where each constraint $i$ is represented by $k_i$ of these vertices.
- $R$ is the set of vertices in the other side of the bipartite graph. It contains a vertex for each item $c \in C$ and for each dummy item $c' \in C_{\text{dummy}}$.
- $E$ is the set of edges. Each vertex in $R$ is connected to each vertex of each of the constraints that it satisfies.
- The weight $w(l, r)$ of edge $(l, r) \in E$ is $v_l(r)$: the value of item $r$ associated with property $P_l$.

There is a natural correspondence between *saturated-matchings* in this graph, that is matchings in which every $l \in L$ is matched, and between *feasible solutions* (i.e., solutions that satisfy the constraints) to the allocation problem. Thus, a saturated-matching of maximum value corresponds to an optimal solution. It is well know that the problem of finding such a maximum weight bipartite matching can be solved in polynomial time (see e.g., Lawler, 2001).

**Problem definition.** In this work we consider the following online learning model. We assume that $n$ items are sequentially drawn i.i.d. from an unknown distribution $D$ over $X$. Upon receiving each item, we decide whether to retain it, or reject it irrevocably (the first stage of the algorithm). Thereafter, we select a feasible solution[6] consisting *only* of retained items (the second stage of the algorithm). Most importantly, before accessing the online sequence and take irreversible online decisions of which items to reject, we have access a training set $C_{\text{train}}$ consisting of $n$ independent draws from $D$.

### 2.1 Results

#### 2.1.1 Oblivious online screening

We begin by studying a greedy algorithm that does not require a training set. In the online phase, this algorithm acts greedily by keeping an item if it participates in the best solution thus far. Then,

the algorithm computes an optimal matching among the retained items. The particular details of the algorithm are given in the supplementary material. We have the following guarantee for this greedy algorithm proven in the supplementary material.

**Theorem 1.** *Let $\delta \in (0,1)$. The greedy algorithm outputs the optimal solution with probability at least $1 - \delta$ and retains $O(k \log(\min\{k/\delta, n/k\}))$ items in expectation.*

As we shall see in the next section, learning from the training set allows one to retain exponentially less items than is implied by the theorem above.[7] It is then natural to ask to which extent is the training phase essential in order to accommodate such an improvement. We answer this question in The supplementary material by proving a lower bound on the number of retained items for *any* algorithm that does not use a training phase. This lower bound already applies in the simple setting where $d = 1$: here, each item consists only of a value $v \in [0, 1]$, and the goal of the algorithm is to retain as few items as possible while guaranteeing with high probability that the top $k$ maximal values are retained.

**Theorem 2.** *Let $\delta \in (0,1)$. For every algorithm A which retains the maximal $k$ elements with probability at least $1 - \delta$, there exists a distribution $\mu$ such that the expected number of retained elements for input sequences $v_1 \ldots v_n \sim \mu^n$ is at least $\Omega(k \log(\min\{k/\delta, n/k\}))$.*

Thus, the above theorem implies that $\Theta(k \log(n/k))$ can not be improved even if we allow failure probability $\delta = \Theta(k^2/n)$ (see Theorem 1).

### 2.1.2 Online screening with learning

We now design online algorithms that, before the online screening process begins, use $C_{\text{train}}$ to learn a *thresholds-policy* $T \in \mathcal{T}$ such that with high probability: (i) the number of items that are retained in the online phase is small, and (ii) there is a feasible solution consisting of $k$ retained items whose value is optimal (or close to optimal). Thresholds-policies are studied in Section 3 and are defined as follows.

**Definition 3** (Thresholds-policies)**.** A threshold-policy is parametrized by a vector $T = (t_1, \ldots, t_d)$ of thresholds, where $t_i$ corresponds to property $P_i$ for $1 \leq i \leq d$. The semantics of $T$ is as follows: given a sample $C$ of $n$ items, each item $c \in C$ is retained if and only if there exists a property $P_i$ satisfied by $c$, such that its value $v_i(c)$ passes the threshold $t_i$. More formally, $c$ is retained if and only if $\exists i \in \{1, \ldots, d\}$ such that $c \in P_i$ and $v_i(c) \geq t_i$.

Having proven uniform convergence results for thresholds-policies (see Section 3.1), we show the following in Section 4.

**Theorem 4.** *There exists an algorithm that learns a thresholds-policy T from a single training sample $C_{\text{train}} \sim D^n$, such that after processing the ("real-time") input sample $C \sim D^n$ using T:*

- *It outputs an optimal solution with probability at least $1 - \delta$.*

- *The expected number of retained items in the first phase is $O\big(k(\log d + \log\log(n/k) + \log\log(1/\delta))\big)$.*

Thus, with the additional information given by the training set, the algorithm presented in Theorem 4 improves the number of retained items from $k \log(k/\delta)$ to $k \log\log(1/\delta)$. This demonstrates a significant improvement over Theorem 1.

Finally, in the supplementary material we prove that the algorithm from Theorem 4 is nearly-optimal in the sense that it is impossible to significantly improve the number of retained items even if we allow the algorithm to fully know the distribution over input items (so, in a sense, having an access to $n$ i.i.d samples from the distribution is the same as knowing it completely).

**Theorem 5.** *Consider the case where $k = d$ and $k_1 = \cdots k_d = 1$. There exists a universe X and a fixed distribution D over X such that for $C \sim D^n$ the following holds: any online learning algorithm (which possibly "knows" D) that retains a subset $S \subseteq C$ of items that contains an optimal solution with probability at least $1 - \delta$ must satisfy that $\mathsf{Ex}\big[|S|\big] = \Omega(k \log\log(1/\delta))$.*

# 3 Thresholds-policies

We next discuss a framework to design algorithms that exploit the training set to learn policies that are applied in the first phase of the matching process. We would like to frame this in standard ML formalism by phrasing this problem as learning a class $\mathcal{H}$ of policies such that:

- $\mathcal{H}$ **is not too small:** The policies in $\mathcal{H}$ should yield solutions with high values (optimal, or near-optimal).

- $\mathcal{H}$ **is not too large:** $\mathcal{H}$ should satisfy some uniform convergence properties; i.e. the performance of each policy in $\mathcal{H}$ on the training set is close, with high probability, to its expected real-time performance on the sampled items during the online selection process.

Indeed, as we now show these demands are met by the class $\mathcal{T}$ of thresholds policies (Definition 3). We first show that the class of thresholds-policies contains an optimal policy, and in the sequel we show that it satisfies attractive uniform convergence properties.

**An assumption (values are unique).** We assume that for each constraint $P_i$, the marginal distribution over the value of $c \sim D$ conditioned on $c \in P_i$ is atomless; namely $\Pr_{c \sim D}[v(c) = v \mid c \in P_i] = 0$ for every $v \in [0, 1]$. This assumption can be removed by adding artificial tie-breaking rules, but making it will simplify some of the technical statements.

**Theorem 6** (There is a thresholds policy that retains an optimal solution)**.** *For any set of items $C$, there exists a thresholds vector $T \in \mathcal{T}$ that retains exactly $k$ items that form an optimal solution for $C$.*

*Proof.* Let $S$ denote the set of $k$ items in an optimal solution for $C$, and let $S_i \subseteq S \cap P_i$ be the subset of $M$ that is assigned to the constraint $P_i$. Define $t_i = \min_{c \in S_i} v_i(c)$, for $i \geq 1$, Clearly, $T$ retains all the items in $S$. Assume towards contradiction that $T$ retains an item $c_j \notin S$, and assume that $P_i$ is a constraint such that $c_j \in P_i$ and $v_i(c_j) \geq t_i$. Since by our assumption on $D$ all the values $v_i(c_j)$ are distinct it follows that $v_i(c_j) > t_i$. Thus, we can modify $S$ by replacing $c_j$ with the item of minimum value in $S_i$ and increase the total value. This contradicts the optimality of $S$. $\qquad\square$

We next establish generalization bounds for the class of thresholds-policies.

## 3.1 Uniform convergence of the number of retained items

For a sample $C \sim D^n$ and a thresholds-policy $T \in \mathcal{T}$, we denote by $R_i^T(C) = \{c : c \in P_i \text{ and } v_i(c) \geq t_i\}$ the set of items that are retained by the threshold $t_i$, and we denote its expected size by $\rho_i^T = \mathsf{Ex}_{C \sim D^n}\left[|R_i^T(C)|\right]$. Similarly we denote by $R^T(C) = \cup_i R_i^T(C)$ the items retained by $T$, and by $\rho^T$ its expectation. We prove that the sizes of $R_i^T(C)$ and $R^T(C)$ are concentrated around their expectations uniformly for all thresholds policies.

The following theorems establish uniform convergence results for the number of retained items. Namely, with high probability we have $R_i^T \approx \rho_i^T$, $R^T \approx \rho^T$ simultaneously for all $T \in \mathcal{T}$ and $i \leq d$.

**Theorem 7** (Uniform convergence of the number of retained items)**.** *With probability at least $1 - \delta$ over $C \sim D^n$, the following holds for all policies $T \in \mathcal{T}$ simultaneously:*

1. *If $\rho^T \geq k$, then $(1 - \varepsilon)\rho^T \leq |R^T(C)| \leq (1 + \varepsilon)\rho^T$, and*

2. *if $\rho^T < k$, then $\rho^T - \varepsilon k \leq |R^T(C)| \leq \rho^T + \varepsilon k$,*

*where*

$$\varepsilon = O\left(\sqrt{\frac{d \log(d) \log(n/k) + \log(1/\delta)}{k}}\right).$$

**Theorem 8** (Uniform convergence of the number of retained items per constraint)**.** *With probability at least $1 - \delta$ over $C \sim D^n$, the following holds for all policies $T \in \mathcal{T}$ and all $i \leq d + 1$ simultaneously:*

1. *If $\rho_i^T \geq k$, then $(1 - \varepsilon)\rho_i^T \leq |R_i^T(C)| \leq (1 + \varepsilon)\rho_i^T$, and*

2. *if $\rho_i^T < k$, then $\rho_i^T - \varepsilon k \leq |R_i^T(C)| \leq \rho_i^T + \varepsilon k$,*

*where*

$$\varepsilon = O\left(\sqrt{\frac{\log(d)\log(n/k) + \log(1/\delta)}{k}}\right).$$

The proofs of Theorems 7 and 8 are based on standard VC-based uniform convergence results, and technically the proof boils down to bounding the VC-dimension of the families

$$\mathcal{R} = \{R^T : T \in \mathcal{T}\} \quad \text{and} \quad \mathcal{Q} = \{R_i^T : T \in \mathcal{T}, \ i \leq d\}.$$

Indeed, in the supplementary material we prove the following.

**Lemma 9.** $VC(\mathcal{R}) = O(d \log d)$.

**Lemma 10.** $VC(\mathcal{Q}) = O(\log d)$.

Using Lemmas 9 and 10, we can now apply standard uniform convergence results from VC-theory to derive Theorems 7 and 8.

**Definition 11** (Relative $(p, \varepsilon)$-approximation; Har-Peled and Sharir, 2011)**.** Let $\mathcal{F}$ be a family of subsets over a domain $X$, and let $\mu$ be a distribution on $X$. $Z \subseteq X$ is a $(p, \varepsilon)$-approximation for $\mathcal{F}$ if for each $f \in F$ we have,

1. If $\mu(f) \geq p$, then $(1 - \varepsilon)\mu(f) \leq \widehat{\mu}(f) \leq (1 + \varepsilon)\mu(f)$,

2. If $\mu(f) < p$, then $\mu(f) - \varepsilon p \leq \widehat{\mu}(f) \leq \mu(f) + \varepsilon p$,

where $\widehat{\mu}(f) = |Z \cap F|/|Z|$ is the ("empirical") measure of $f$ with respect to $Z$.

The proof of Theorems 7 and 8 now follows by plugging $p = k/n$ in Har-Peled and Sharir [2011, Theorem 2.11], which we state in the next proposition.

**Proposition 12** (Har-Peled and Sharir, 2011)**.** *Let $\mathcal{F}$ and $\mu$ like in Definition 11. Suppose $\mathcal{F}$ has VC dimension $m$. Then, with provability at least $1 - \delta$, a random sample of size*

$$\Omega\left(\frac{m\log(1/p) + \log(1/\delta)}{\varepsilon^2 p}\right)$$

*is a relative $(p, \varepsilon)$-approximation for $\mathcal{F}$.*

### 3.2 Uniform convergence of values

We now prove a concentration result for the value of an optimal solution among the retained items. Unlike the number of retained items, the value of an optimal solution corresponds to a more complex random variable, and analyzing the concentration of its empirical estimate requires more advanced techniques.

We denote by $V^T(C)$ the value of the optimal solution among the items retained by the thresholds-policy $T$, and we denote its expectation by $\nu^T = \text{Ex}_{C \sim D^n}\left[V^T(C)\right]$. We show that $V^T(C)$ is concentrated uniformly for all thresholds policies.

**Theorem 13** (Uniform convergence of values)**.** *With probability at least $1 - \delta$ over $C \sim D^n$, the following holds for all policies $T \in \mathcal{T}$ simultaneously:*

$$|\nu^T - V^T(C)| \leq \varepsilon k, \quad \text{where} \ \varepsilon = O\left(\sqrt{\frac{d\log k + \log(1/\delta)}{k}}\right).$$

Note that unlike most uniform convergence results that guarantee simultaneous convergence of empirical averages to expectations, here $V^T(C)$ is not an average of the $n$ samples, but rather a more complicated function of them. We also note that a bound of $\widetilde{O}(\sqrt{n})$ (rather than $\widetilde{O}(\sqrt{k})$) on the additive deviation of $V^T(C)$ from its expectation can be derived using the McDiarmid's inequality [McDiarmid, 1989]. However, this bound is meaningless when $\sqrt{n} > k$ (because $k$ upper bounds the value of the optimal solution). We use Talagrand's concentration inequality [Talagrand, 1995] to derive the $O(\sqrt{k})$ upper bound on the additive deviation. Talagrand's concentration inequality allows us to utilize the fact that an optimal solution uses only $k \ll n$ items, and therefore replacing an item that does not participate in the solution does not affect its value.

To prove the theorem we need the following concentration inequality for the value of the optimal selection in hindsight. Note that by Theorem 6 this value equals to $V^T(C)$ for some $T$.

**Lemma 14.** *Let* $\mathsf{OPT}(C)$ *denote the value of the optimal solution for a sample C. We have that*

$$\Pr_{C \sim D^n} \left[ |\mathsf{OPT}(C) - \mathsf{Ex}[\mathsf{OPT}(C)]| \geq \alpha \right] \leq 2\exp(-\alpha^2/2k).$$

So, for example, it happens that $|\mathsf{OPT}(C) - \mathsf{Ex}[\mathsf{OPT}(C)]| \leq \sqrt{2k\log(2/\delta)}$ with probability at least $1 - \delta$.

To prove this lemma we use the following version of Talagrand's inequality (that appears for example in lecture notes by van Handel [2014]).

**Proposition 15** (Talagrand's Concentration Inequality). *Let* $f : \mathbb{R}^n \mapsto \mathbb{R}$ *be a function, and suppose that there exist* $g_1, \ldots, g_n : \mathbb{R}^n \mapsto \mathbb{R}$ *such that for any* $x, y \in \mathbb{R}^n$

$$f(x) - f(y) \leq \sum_{i=1}^{n} g_i(x) 1_{[x_i \neq y_i]}. \tag{1}$$

*Then, for independent random variables* $X = (X_1, \ldots, X_n)$ *we have*

$$\Pr\left[ |f(X) - \mathsf{Ex}[f(X)]| > \alpha \right] \leq 2\exp\left( -\frac{\alpha^2}{2\sup_x \sum_{i=1}^{n} g_i^2(x)} \right).$$

*Proof of Lemma 14.* We apply Talagrand's concentration inequality to the random variable $\mathsf{OPT}(C)$. Our $X_i$'s are the items $c_1, \ldots, c_n$ in the order that they are given. We show that Eq. (1) holds for $g_i(C) = 1_{[c_i \in S]}$ where $S = S(C)$ is a fixed optimal solution for $C$ (we use some arbitrary tie breaking among optimal solutions). We then have, $\sum_{i=1}^{n} g_i^2(C) = |S| = k$, thus completing the proof.

Now, let $C$, $C'$ be two samples of $n$ items. Recall that we need to show that

$$\mathsf{OPT}(C) - \mathsf{OPT}(C') \leq \sum_{i=1}^{n} g_i(C) 1_{[c_i \neq c_i']}.$$

We use $S$ to construct a solution $S'$ for $C'$ as follows. Let $S_j \subseteq S$ the subset of $S$ matched to $P_j$. For each $i$, if $c_i \in S_j$ for some $j$, and $c_i = c_i'$, then we add $i$ to $S_j'$. Otherwise, we add a dummy item from $C'_{\text{dummy}}$ to $S_j'$ (with value zero). Let $V(S')$ denote the value of $S'$. Note that the difference between the values of $S$ and $S'$ is the total value of all items $i \in S$ such that $c_i \neq c_i'$. Since the item values are bounded in $[0, 1]$ we get that

$$\mathsf{OPT}(C) - V(S') = \sum_{j=1}^{d} \sum_{c_i \in S_j} v_j(c_i) 1_{[c_i \neq c_i']} \leq \sum_{j=1}^{d} \sum_{c_i \in S_j} 1_{[c_i \neq c_i']} = \sum_{i=1}^{n} g_i(C) 1_{[c_i \neq c_i']}.$$

The proof is complete by noticing that $\mathsf{OPT}(C') \geq V(S')$. $\qquad\square$

We also require the following construction of a bracketing of $\mathcal{T}$ which is formally presented in the supplementary material.

**Lemma 16.** *There exists a collection of* $\mathcal{N}$ *thresholds-policies such that* $|\mathcal{N}| \leq k^{O(d)}$, *and for every thresholds-policy* $T \in \mathcal{T}$ *there are* $T^+, T^- \in \mathcal{N}$ *such that*

1. $V^{T^-}(C) \leq V^T(C) \leq V^{T^+}(C)$ *for every sample of items C; note that by taking expectations this implies that* $v^{T^-} \leq v^T \leq v^{T^+}$, *and*

2. $v^{T^+} - v^{T^-} \leq 10$.

*Proof of Theorem 13.* The items in $C$ that are retained by $T$ are independent samples from a distribution $D'$ that is sampled as follows: (i) sample $c \sim D$, and (ii) if $c$ is retained by $T$ then keep it, and otherwise discard it. This means that $v^T(C)$ is in fact the optimal solution of $C$ with respect to $D'$. Since Lemma 14 applies to *every* distribution $D$ we can apply it to $D'$ and get that for any fixed $T \in \mathcal{T}$

$$\Pr_{C \sim D^n} \left[ |v^T - V^T(C)| \geq \alpha \right] \leq 2\exp(-\alpha^2/2k).$$

Now, by the union bound for $\mathcal{N}$ be as in Lemma 16 we get that the probability that there is $T \in \mathcal{N}$ such that $|v^T - V^T(C)| \geq \alpha$ is at most $|\mathcal{N}| \cdot 2\exp(-\alpha^2/2k)$. Thus, since $|\mathcal{N}| \leq k^{O(d)}$, it follows that with probability at least $1 - \delta$,

$$(\forall T \in \mathcal{N}): \ |v^T - V^T(C)| \leq O\left(\sqrt{k\left(d\log k + \log(1/\delta)\right)}\right). \tag{2}$$

We now show why uniform convergence for $\mathcal{N}$ implies uniform convergence for $\mathcal{T}$. Combining Lemma 16 with Equation (2) we get that with probability at least $1 - \delta$, every $T \in \mathcal{T}$ satisfies:

$$
\begin{aligned}
|v^T - V^T(C)| &\leq \max\{|v^{T^+} - V^{T^-}(C)|, |v^{T^-} - V^{T^+}(C)|\} &\text{(by Item 1 of Lemma 16)}\\
&\leq \max\{|v^{T^-} - V^{T^-}(C)|, |v^{T^+} - V^{T^+}(C)|\} + 10 &\text{(by Item 2 of Lemma 16)}\\
&\leq 10 + O\left(\sqrt{k\left(d\log k + \log(1/\delta)\right)}\right). &\text{(by Eq. (2))}
\end{aligned}
$$

Here the first inequality follows from Item 1 by noticing that if $[a,b], [c,d]$ are intervals on the real line and $x \in [a,b], y \in [c,d]$ then $|x-y| \leq \max\{|b-c|, |d-a|\}$, and plugging in $x = v^T, y = V^T(C), a = v^{T^-}, b = v^{T^+}, c = V^{T^-}(C), d = V^{T^+}(C)$.

This finishes the proof, by setting $\varepsilon$ such that $\varepsilon \cdot k = O\left(\sqrt{k(d\log k + \log(1/\delta))}\right)$. □

## 4   Algorithms based on learning thresholds-policies

We next exemplify how one can use the above properties of thresholds-policies to design algorithms. A natural algorithm would be to use the training set to learn a threshold-policy $T$ that retains an optimal solution with $k$ items from the training set as specified in Theorem 6, and then use this online policy to retain a subset of the $n$ items in the first phase. Theorem 7 and Theorem 13 imply that with probability $1 - \delta$, the number of retained items is at most $m = k + O\left(\sqrt{kd\log(d)\log(n/k) + k\log(1/\delta)}\right)$ and that the value of the resulting solution is at least $\mathsf{OPT} - O\left(\sqrt{kd\log k + k\log(1/\delta)}\right)$.

We can improve this algorithm by combining it with the greedy algorithm of Theorem 1 described in the supplementary material. During the first phase, we retain an item $c$ only if (i) $c$ is retained by $T$, and (ii) $c$ participates in the optimal solution among the items that were retained thus far. Theorem 1 then implies that out of these $m$ items greedy keeps a subset of

$$O\left(k\log\frac{m}{k}\right) = O\left(k\left(\log\log\left(\frac{n}{k}\right) + \log\log\left(\frac{1}{\delta}\right)\right)\right).$$

items in expectation that still contains a solution of value at least $\mathsf{OPT} - O(\sqrt{kd\log k + k\log(1/\delta)})$.

We can further improve the value of the solution and guarantee that it will be optimal (with respect to all $n$ items) with probability $1 - \delta$. This is based on the observation that if the set of retained items contains the top $k$ items of each property $P_i$ then it also contains an optimal solution. Thus, we can compute a thresholds-policy $T$ that retains the top $k + O(\sqrt{k\log(d)\log(n/k) + k\log(1/\delta)})$ items of each property from the training set (if the training set does not have this many items with some property then set the corresponding threshold to 0). Then, it follows from Theorem 8, that with probability $1 - \delta$, $T$ will retain the top $k$ items of each property in the first online phase and therefore will retain an optimal solution. Now, Theorem 8 implies that with probability $1 - \delta$ the total number of items that are retained by $T$ in real-time is at most $m = dk + O(d\sqrt{k\log(d)\log(n/k) + k\log(1/\delta)})$. By filtering the retained elements with the greedy algorithm of Theorem 1 as before it follows that the total number of retained items is at most

$$k + k\log\left(\frac{m}{k}\right) = O\left(k\left(\log d + \log\log\left(\frac{n}{k}\right) + \log\log\left(\frac{1}{\delta}\right)\right)\right)$$

with probably $1 - \delta$. This proves Theorem 4.

## Acknowledgements

We thank an anonymous reviewer for their remarks regarding a previous version of this manuscript. Their remarks and questions eventually led us to proving Theorem 2.

## Footnotes

[6]In addition to the retained items, the algorithm has access to $C_{\text{dummy}}$, and therefore a feasible solution always exists.

[7]That is, the expected number of retained items is reduces from order of $\log n$ to $\log\log n$.

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
