[Supplementary Material]

# Learning to Screen

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

[2] We assume $\delta n/k$ is an integer without loss of generality.

[3]Such $t_i^j$'s exist due to our assumption that $D$ is atomless (see Section 3).

[4]When $A$ is randomized then the value of $\Pr[A \text{ retains } v_k]$ depends only on the order-type of $v_1 \ldots v_m$.

[5]Note that to apply Lemma 21 on $S_i$ we need $\delta > e^{-n_i/2}$, which is equivalent to $n_i > 2\ln(1/\delta)$.

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

## A Deferred Proofs

### A.1 The Greedy Online Algorithm

A simple way to collect a small set of items that contains the optimal solution is to select the $k$ largest items of each property. This set clearly contains the optimal solution. A simple argument, as in the proof of Lemma 18, shows that this implementation of the first stage keeps $O(kd \log(n/k))$ items on average. In the following we present a *greedy algorithm* that retains an average number of $O(k \log(k/\delta))$ items in the first phase (for a parameter $\delta \in (0, 1)$).

The greedy algorithm works as follows: it ignores the first $\delta n/k$ items[2] and then starts processing the items one by one. When we process the $i$'th item, $c_i$, the algorithm computes the optimal solution $M_i$ of the first $i$ items (recall that we assume the algorithm has access to $C_{\text{dummy}}$, a large enough pool of zero valued items so there is always a feasible solution). The greedy algorithm retains $c_i$ if and only if $c_i$ participates in $M_i$. We assume that $M_i$ is unique for every $i$ (we can achieve this with an arbitrary consistent tie breaking rule, say among matchings of the same value we prefer the one that maximizes the sum of the indices of the matched items.). Since the optimal solutions correspond to maximum-weighted bipartite-matchings between the items and the constraints, we have the following lemma.

**Lemma 17.** *Suppose that the optimal solution, denoted by $M$, does not appear before round $\delta n/k$. Then it is a subset of the retained items.*

*Proof.* Let $i \geq \delta n/k$. Consider an item $c$ matched by $M$ and assume by contradiction that $c$ is not matched in $M_i$. Consider $Z = M \triangle M_i$ (we take the symmetric difference of $M$ and $M_i$ as sets of edges). Since $M$ and $M_i$ do not necessarily match the same items then the edges in $Z$ induce a collection of alternating paths and cycles where each path $L$ has an item matched by $M$ and not by $M_i$ at one end, and an item matched by $M_i$ and not by $M$ at the other hand. Except for its two ends, an alternating path contains items that are matched by both $M$ and $M_i$. From the optimality and the uniqueness of $M$ follows that for each path the value of $M$ is larger than the value of $M_i$.

Since $c$ is matched by $M$ and not by $M_i$ there is a path $L$ in $Z$ that starts at $c$ and ends at some item that is matched by $M_i$ and not by $M$.

It follows that all the items in $L$ are in $M_i$ and if we match them according to $M$ then the value that we gain from them increases. This contradicts the optimality of $M_i$.

(Note that, in fact, there are no cycles in $Z$, since they will imply that there are multiple optimal solutions, contradicting the uniqueness of $M_i$ and $M$.) □

Lemma 17 implies that, with high probability, if we collect all items that are in the optimal solution of the subset of items that precedes them then the set of items that we have at the end contains the optimal solution. Indeed, our algorithm fails if at least one of the items in the optimal solution $M$ is among the first $\delta n/k$ items. The probability that this occurs is at most $\delta$ via a union bound and the fact that the probability that any fixed item in $M$ is among the first $\delta n/k$ items is exactly $\delta/k$.

The next question is: how large is the subset of the items which we retain? The next lemma answers this question in an average sense.

**Lemma 18.** *Assume that at the first stage the algorithm receives the items in a random order. Then the expected number of items that the first stage keeps is $O\left(k \log \min \left\{ \frac{n}{k}, \frac{k}{\delta} \right\}\right)$.*

*Proof.* Let $i \geq \delta n/k$ and denote $X_i$ as an indicator that is one if and only if the $i$'th item belongs to $M_i$. Condition the probability space on the *set* $L_i$ of the first $i$ items (but not on their order). Each element of $L_i$ is equally likely to arrive last. So since $|M_i| \leq k$, then the probability that the element arriving last in $L_i$ is in $\mathsf{OPT}_i$ is at most $k/i$ if $k < i$ or at most 1 otherwise. It follows that $E[X_i \mid L_i] \leq \min\left\{\frac{k}{i}, 1\right\}$. Since this holds for any $L_i$, it also holds unconditionally as well. Therefore, if $\delta n/k < k$ then by the fact that $\sum_{i=k+1}^{n} \frac{1}{i} \leq \log \frac{n}{k}$, the expected number of retained items is

$$k - \frac{\delta n}{k} + \sum_{i=k+1}^{n} \frac{k}{i} = O\left(k \log \frac{n}{k}\right).$$

414   Similarly, if $\delta n/k \geq k$ then the expected number of retained items is

$$\sum_{i=\delta n/k+1}^{n} \frac{k}{i} = O\left(k \log \frac{k}{\delta}\right). \qquad \square$$

## A.2   Generalization and concentration

415

416   **Technical notation.**   For $m \in \mathbb{N}$, the set $\{1, \ldots, m\}$ is denoted by $[m]$. Given a family of sets $F$ over
417   a domain $X$, and $Y \subseteq X$, the family $\{f \cap Y : f \in F\}$ is denoted by $F|_Y$. Recall that the VC dimension
418   of $F$ is the maximum size of $Y \subseteq X$ such that $F|_Y$ contains all subsets of $Y$.

419   **Lemma** (restatement of Lemma 9).  $VC(\mathcal{R}) = O(d \log d)$ .

420   *Proof.*  Let $S$ be a set of items shattered by $\mathcal{R}$ and denote its size by $m$; since $S$ is arbitrary, an upper
421   bound on $m$ implies an upper bound on $VC(\mathcal{R})$. To this end we upper bound the number of subsets
422   in $\mathcal{R}|_S = \{S \cap R_T : R_T \in \mathcal{R}\}$. Now, there are $m$ items in $S$ with at most $m$ different values. Therefore,
423   we can restrict our attention to thresholds-policies where each threshold is picked from a fixed set of
424   $m+1$ meaningful locations (one location in between values of two consecutive items when we sort the
425   items by value). Thus $|\mathcal{R}|_S| \leq (m+1)^d$, but, as $S$ is shattered, $|\mathcal{R}|_S| = 2^m$ and we get $m \leq d \log_2(m+1)$.
426   This implies $m = O(d \log d)$ from which we conclude that $VC(\mathcal{R}) = O(d \log d)$. $\qquad \square$

427   **Lemma** (restatement of Lemma 10).  $VC(\mathcal{Q}) = O(\log d)$ .

428   *Proof.*  For $i \leq d$, let $\mathcal{Q}_i = \{R_i^T : T \in \mathcal{T}\}$. Note that $\mathcal{Q} = \cup_i \mathcal{Q}_i$. We claim that $VC(\mathcal{Q}_i) = 1$ for all $i$.
429   Indeed, let $c', c''$ be two items. Note that if $c' \notin P_i$ or $c'' \notin P_i$ then $\{c', c''\}$ is not contained by $\mathcal{Q}_i$ and
430   therefore not shattered by it. Therefore, assume that $c', c'' \in P_i$ and $v_i(c') \geq v_i(c'')$. Now, it follows
431   that any threshold $T$ that retains $c''$ must also retain $c'$, and so it follows that also in this case $\{c', c''\}$
432   is not shattered.

433   The bound on the VC dimension of $\mathcal{Q} = \cup_{i \leq d} \mathcal{Q}_i$ follows from the next lemma.

434   **Lemma 19.**  *Let $m \geq 2$ and let $F_1, \ldots, F_m$ be classes with VC dimension at most* 1*. Then, the VC*
435   *dimension of $\cup_i F_i$ is at most* $10 \log m$.

436   *Proof.*  We show that $\cup_i F_i$ does not shatter a set of size $10 \log m$. Let $Y \subseteq X$ of size $10 \log m$. Indeed,
437   by the Sauer's Lemma [Sauer, 1972]:

$$|(\cup_i F_i)|_Y| \leq m\left(\binom{10 \log m}{0} + \binom{10 \log m}{1}\right) = m(1 + 10 \log m) < m^{10} = 2^{10 \log m},$$

438   and therefore, $Y$ is not shattered by $\cup_i F_i$. $\qquad \square$

439   This finishes the proof of Lemma 10. $\qquad \square$

440   **Lemma** (restatement of Lemma 16).  *There exists a collection of $\mathcal{N}$ thresholds-policies such that*
441   *$|\mathcal{N}| \leq k^{O(d)}$, and for every thresholds-policy $T \in \mathcal{T}$ there are $T^+, T^- \in \mathcal{N}$ such that*

442      *1.  $V^{T^-}(C) \leq V^T(C) \leq V^{T^+}(C)$ for every sample of items C. (By taking expectations this also*
443         *implies that $\nu^{T^-} \leq \nu^T \leq \nu^{T^+}$.)*

444      *2.  $\nu^{T^+} - \nu^{T^-} \leq 10$.*

445   *Proof.*  For every $i \leq d$ and $j \leq dn$ define thresholds $t_i^j \in [0, 1]$ where $t_i^0 = 1$ and for $j > 0$ set $t_i^j$ to
446   satisfy[3]

$$\Pr_{c \sim D}[v(c) \geq t_i^j \text{ and } c \in P_i] = \frac{j}{dn}.$$

447   Note that $t_i^0 > t_i^1 > \ldots$ (see Figure 1). Set

$$\mathcal{J}_i = \left\{j : 0 \leq \frac{j}{dn} \leq \Pr_{c \sim D}[c \in P_i], j \in \mathbb{N}\right\},$$

Figure 1: An illustration of the thresholds in $\mathcal{N}_i$ as defined in the proof of Lemma 16. Each $t_i^j$ for $j \in \mathcal{J}_i$ satisfies $\Pr_{c \sim D}[v(c) \geq t_i^j$ and $c \in P_i] = \frac{j}{dn}$.

448  and define

$$\mathcal{N}_i = \{t_i^j \mid j \in \mathcal{J}_i \cap \{0, 1, \ldots, 10dk\}\} \cup \{0\}$$
$$\mathcal{N} = \mathcal{N}_1 \times \mathcal{N}_2 \ldots \times \mathcal{N}_d.$$

449  Note that indeed $|\mathcal{N}| \leq (10dk + 2)^{d+1} = k^{O(d)}$.

450  We next show that $\mathcal{N}$ satisfies items 1 and 2 in the statement of the lemma. Let $T \in \mathcal{T}$ be an arbitrary
451  thresholds-policy. The policies $T^- = (t_i^-)_{i \leq d}$, and $T^+ = (t_i^+)_{i \leq d}$ are derived by rounding $t$ in each
452  coordinate up and down respectively, to the closest policies in $\overline{\mathcal{N}}$ (so, the thresholds in $T^+$ are smaller
453  than in $T^-$; the "+" sign reflects that it retains more items and achieves a higher value). Formally,
454  $t_i^+ = \max\{t \in \mathcal{N}_i : t \leq t_i\}$ and $t_i^- = \min\{t \in \mathcal{N}_i : t \geq t_i\}$ where $t_i$ is the threshold for property $i$ in
455  $T$. Therefore, for every sample $C \sim D^n$, the set of items in $C$ that are retained by $T$ contains the set
456  retained by $T^-$ and is contained in the set retained by $T^+$. This implies item 1.

457  To derive item 2, observe that for every sample $C$: $V^{T^+}(C) - V^{T^-}(C) \leq |Z|$, where $Z \subseteq C$ denotes
458  the set of items which participate in some canonical optimal solution for $T^+$ that are not retained by
459  $T^-$. Thus, it suffices to show that $\mathsf{Ex}[|Z|] \leq 10$. To this end put $p_i = \Pr_{c \sim D}[v(c) \geq t_i$ and $c \in P_i]$ and
460  partition $Z$ into two disjoint sets $Z = E \cup F$, where $E$ is the set of all items $c_j \in Z$ that are assigned by
461  the optimal solution of $T^+$ to a property $P_i$ where $p_i < \frac{10k}{n}$, and $F = Z \setminus E$. We claim that

462  • $\mathsf{Ex}[|E|] \leq 1$: for each $P_i$ such that $p_i < \frac{10k}{n}$ let $G_i \subseteq P_i$ denote the set of items whose
463    value $v \in [t_i^+, t_i^-)$ (i.e. retained by $T^+$ and not by $T^-$). Note that $E \subseteq \cup_i G_i$, and that
464    $\Pr_{c \sim D}[c \in G_i] \leq \frac{1}{dn}$. Thus, it follows that

$$\mathop{\mathsf{Ex}}_{C \sim D^n}[|E|] \leq \mathop{\mathsf{Ex}}_{C \sim D^n}[|\cup_i G_i|] \leq \sum_i \mathop{\mathsf{Ex}}_{C \sim D^n}[|G_i|] \leq d \cdot \frac{n}{dn} \leq 1.$$

465  • $\mathsf{Ex}[|F|] \leq 9$: note that $\mathsf{Ex}[|F|] \leq k \cdot \Pr[|F| > 0]$ (because $F \subseteq Z$ and $|Z| \leq k$). Thus,
466    it suffices to show that $\Pr[F > 0] \leq \frac{9}{k}$. Indeed, $F \neq \emptyset$ only if there is a property $P_i$ with
467    $p_i \geq \frac{10k}{n}$ such that less than $k$ items from $P_i$ are retained by $T^-$. Fix a property $P_i$ such that
468    $p_i \geq \frac{10k}{n}$ and let $p_i^- = \Pr_{c \sim D}[v(c) \geq t_i^-$ and $c \in P_i]$. Since $p_i^- \geq \frac{10k}{n}$, a multiplicative Chernoff
469    bound yields that

$$\mathop{\Pr}_{C \sim D^n}[\text{less than } k \text{ items from } P_i \text{ are retained by } T^-] \leq \exp\left(-\frac{(9/10)^2}{2}10k\right) \leq \frac{9}{k^2} \leq \frac{9}{dk},$$

470    and a union bound over all such properties $P_i$ implies that $\Pr[|F| > 0] \leq \frac{9d}{dk} \leq \frac{9}{k}$.

471  Thus, it follows that $v^{T^+} - v^{T^-} \leq 1 + k \cdot \frac{9}{k} = 10$, which finishes the proof.

472  $\qquad\qquad\qquad\qquad\qquad\qquad\qquad\qquad\qquad\qquad\qquad\qquad\qquad\qquad\qquad\qquad\qquad\qquad\qquad\quad$ $\square$

## B Lower Bounds

### B.1 Necessity of the training phase

Let $n \in \mathbb{N}$ (sample size) and $\delta \in [0, 1]$ (confidence parameter). In this section we focus on the case where there is no training phase and $d = 1$. Thus, we consider algorithms which get as an input a sequence $v_1, \ldots v_n \in [0, 1]$ in an online manner (one after the other). In step $m$ the algorithm needs to decide whether to retain $v_m$ or to discard it (this decision may depend on the prefix $v_1 \ldots, v_m$). The algorithm is not allowed to discard a sample after it has been retained.

The following property captures the utility of the algorithm: *for every distribution $\mu$ over $[0, 1]$, if $v_1, \ldots, v_n$ are sampled i.i.d from $\mu$, then with probability at least $1-\delta$, the algorithm retains $v_{j_1}, \ldots, v_{j_k}$ that are the largest $k$ elements in $v_1, \ldots, v_n$*. The goal is to achieve this while minimizing the number of retained items in expectation.

**Theorem** (Theorem 2 restatement). *Let $\delta \in (0, 1)$. For every algorithm A which retains the maximal $k$ elements with probability at least $1 - \delta$, there exists a distribution $\mu$ such that the expected number of retained elements for input sequences $v_1 \ldots v_n \sim \mu^n$ is at least $\Omega(k \log(\min\{n/k, k/\delta\}))$.*

We remind that the bound is tight for the greedy algorithm (Theorem 1).

*Proof.* Following [Moran et al., 1985, Corollary 3.4], we may assume that $A$ accesses its input only using comparisons. More precisely: call two sequences $v_1, \ldots, v_m$ and $u_1, \ldots, u_m$ *order-equivalent* if $v_i \leq v_j \iff u_i \leq u_j$ for all $i, j \leq m$, and call the equivalence class of $v_1 \ldots v_m$ its *order-type*. Note that if $v_1, \ldots, v_m$ are distinct, then their order-type is naturally identified with a permutation $\sigma \in \mathbb{S}_m$. Call an algorithm $A$ *order-invariant* if for every $m \leq n$, the decision[4] of $A$ whether to retain $v_m$ depends only on the order-type of $v_1, \ldots, v_m$ (equivalently, $A$ accesses the input only using comparisons).

By Moran et al. [1985] it follows that for every algorithm $A$ there is an infinite $W \subseteq [0, 1]$ such that $A$ is order-invariant when restricted to input sequences $v_1, \ldots, v_n \in W$. For the remainder of the proof we fix such an infinite set $W$ and focus only on inputs from $W$.

Set $\mu$ to be a uniform distribution over a sufficiently large subset of $W$ so that $v_1 \ldots v_n$ are distinct with probability $1 - 1/n$. Let $\mathsf{OPT}(S)$ denote the top $k$ elements in $S$. Let $T_m$ be the set of all sequences $v_1, \ldots, v_m, \ldots, v_n \in W^n$ such that $v_m \in \mathsf{OPT}(\{v_1, \ldots, v_m\})$, and let $p_k$ denote the probability that $A$ retains $v_m$ conditioned on the input being from $T_m$. Let $T'_m \subseteq T_m$ denote the set of all sequences $v_1, \ldots, v_m, \ldots, v_n$ such that $v_m \in \mathsf{OPT}(\{v_1, \ldots, v_n\})$ (i.e., $v_m$ is part of the optimal solution). The proof hinges on the following lemma:

**Lemma 20.** *Since A is order based, for every $m \leq n$, $p_m$ is also the probability that A retains $v_m$ conditioned on the input being from $T'_m$.*

*Proof.* The decision of $A$ whether to retain $v_m$ depends only on the order-type of $v_1, \ldots, v_m$. For each $\sigma \in \mathbb{S}_m$, let $E(\sigma)$ denote the event that the order type of $v_1 \ldots v_m$ is $\sigma$. Thus,

$$p_m = \Pr[A \text{ retains } v_m \mid T_m] = \sum_{\sigma \in \mathbb{S}_m} \Pr[E(\sigma) \mid T_m] \cdot \Pr[A \text{ retains } v_m \mid E(\sigma)],$$

and similarly

$$\Pr[A \text{ retains } v_m \mid T'_m] = \sum_{\sigma \in \mathbb{S}_m} \Pr[E(\sigma) \mid T'_m] \cdot \Pr[A \text{ retains } v_m \mid E(\sigma)].$$

Next, observe that for each order-type $\sigma \in \mathbb{S}_m$:

$$\Pr[E(\sigma) \mid T_m] = \Pr[E(\sigma) \mid T'_m] = \begin{cases} \frac{1}{m!}, & m \leq k \\ \frac{1}{k(m-1)!}, & v_m \in \mathsf{OPT}(\{v_1, \ldots, v_m\}), m > k \\ 0, & \text{otherwise.} \end{cases} \qquad \square$$

510 With the above lemma in hand, we can finish the proof. For the remainder of the argument, we
511 condition the probability space on the event that all elements in the sequence $v_1, \ldots, v_n$ are distinct
512 and show that conditioned on this event, $A$ retains at least $t = \Omega(\log(1/\delta))$ elements in expectation.
513 Note that this will conclude the proof since by the choice of $\mu$ this event occurs with probability
514 $\geq 1 - 1/n$, which implies that – unconditionally – $A$ retains at least $t - n \cdot (1/n) = t - 1 = \Omega(\log(1/\delta))$
515 elements in expectation.

516 For each $m \leq n$, $v_m$ is among the top $k$ elements with probability $\min\{1, k/m\}$, in which case it is
517 retained with probability $p_m$. So $A$ retains at least

$$\sum_{m=1}^{n} \min\left\{1, \frac{k}{m}\right\} \cdot p_m$$

518 elements in expectation. By the above lemma, the probability that $A$ discards the maximum is

$$\sum_{m=1}^{n} \frac{k}{n} \cdot (1 - p_m),$$

519 which by assumption is smaller than $\delta$. So we obtain that $\sum p_m \geq n(1 - \delta/k)$. Thus, to minimize
520 $\sum_{m=1}^{n} \min\{1, k/m\} \cdot p_m$ subject to the constraint that $\sum p_m \geq n(1 - \delta/k)$ we make the last $n(1 - \delta/k)$
521 $p_m$'s equal to 1 and the rest 0. This gives the desired lower bound. $\qquad \square$

## B.2 The algorithm from Theorem 4 is optimal

523 In the previous section we have presented an algorithm that with probability at least $1 - \delta$ outputs an
524 optimal solution while retaining at most $O(k(\log \log n + \log d + \log \log(1/\delta)))$ items in expectation
525 during the first phase.

526 We now present a proof of Theorem 5. We start with the following lemma that shows the dependence
527 on $\delta$ cannot be improved in general, even for $k = 1$, when there are no constraints, and the distribution
528 over the items is known to the algorithm (so there is no need to train it on a sample from the
529 distribution):

530 **Lemma 21.** *Let $v_1, \ldots, v_n \in [0, 1]$ be drawn uniformly and independently, let $e^{-n/2} < \delta < 1/10$ and*
531 *let $A$ be an algorithm that retains the maximal value among the $v_i$'s with probability at least $1 - \delta$.*
532 *Then,*

$$\mathsf{Ex}\big[|S|\big] = \Omega\left(\log \log \left(\frac{1}{\delta}\right)\right),$$

533 *where $S$ is the set of values retained by the algorithm.*

534 Thus, it follows that for $\delta = \mathsf{poly}(1/n)$ and $k, d = O(1)$ the bound in Theorem 4 is tight.

535 *Proof.* Define $\alpha = \frac{\ln(1/\delta)}{2n} \in (1/n, 1/4)$. Let $E_t$ denote the event that $v_t \geq 1 - \alpha$ and is the largest among
536 $v_1, \ldots, v_t$. We have that

$$\mathsf{Ex}[|S|] \geq \sum_t \Pr[v_t \text{ is picked and } E_t] = \sum_t (\Pr[E_t] - \Pr[v_t \text{ is rejected and } E_t]) \ . \qquad (3)$$

537 We show that since $A$ errs with probability at most $\delta$ then $\sum_t \Pr[E_t \text{ and } v_t \text{ is rejected}]$ is small.

$$\delta \geq \Pr[A \text{ rejects } v_{max}] \geq \sum_t \Pr[A \text{ rejects } v_t \text{ and } E_t \text{ and } v_t = v_{max}]$$

$$= \sum_t \Pr[v_t = v_{max} \mid A \text{ rejects } v_t \text{ and } E_t] \cdot \Pr[A \text{ rejects } v_t \text{ and } E_t]$$

$$\geq \sum_t \Pr[v_i \leq 1 - \alpha \text{ for all } i > t \mid A \text{ rejects } v_t \text{ and } E_t] \cdot \Pr[A \text{ rejects } v_t \text{ and } E_t]$$

$$= \sum_t \Pr[v_i \leq 1 - \alpha \text{ for all } i > t] \cdot \Pr[A \text{ rejects } v_t \text{ and } E_t]$$

$$\geq \sum_t (1-\alpha)^{n-t} \cdot \Pr[A \text{ rejects } v_t \text{ and } E_t]$$

$$\geq (1-\alpha)^n \sum_t \Pr[A \text{ rejects } v_t \text{ and } E_t].$$

The crucial part of the above derivation is in third line. It replaces the event "$v_t = v_{max}$" by the event "$v_i \leq 1 - \alpha$ for all $i > t$" (which is contained in the event "$v_t = v_{max}$" under the above conditioning). The gain is that the events "$v_i \leq 1 - \alpha$ for all $i > t$" and "$A$ rejects $v_t$ and $E_t$" are independent (the first depends only on $v_i$ for $i > t$ and the latter on $v_i$ for $i \leq t$). This justifies the "=" in the fourth line.

Rearranging, we have $\sum_t \Pr[A \text{ rejects } v_t \text{ and } E_t] \leq \frac{\delta}{(1-\alpha)^n}$. Substituting this bound in Eq. (3),

$$\mathsf{Ex}[|S|] \geq \sum_t \Pr[v_t \text{ is picked and } E_t]$$

$$= \sum_t (\Pr[E_t] - \Pr[v_t \text{ is rejected and } E_t])$$

$$= \sum_t \Pr[E_t] - \frac{\delta}{(1-\alpha)^n}$$

$$\geq \frac{1}{4} \ln(\alpha n) - \delta \cdot \exp(2\alpha n) \qquad \text{(explained below)}$$

$$= \frac{1}{4} \ln\left(\frac{\ln(1/\delta)}{2}\right) - \delta \exp(\ln(1/\delta)) \qquad \text{(by the definition of } \alpha)$$

$$= \frac{1}{4} \ln \ln(1/\delta) - \frac{1}{4} \ln 2 - 1 = \Omega(\log\log(1/\delta)),$$

which is what we needed to prove. The last inequality follows because

   (i)  $\sum_t \Pr[E_t] \geq \frac{1}{4} \ln(\alpha n)$ (as is explained next), and

   (ii) $1 - \alpha \geq \exp(-2\alpha)$ for every $\alpha \in [0, \frac{1}{4}]$ (which can be verified using basic analysis).

To see (i), note that

$$\sum_t \Pr[E_t] = \mathsf{Ex}\left[\sum_t 1_{E_t}\right].$$

Let $z = |\{t : v_t \geq 1 - \alpha\}|$. Since the $v_i$'s are uniform in $[0, 1]$ then by the same argument as in the proof of Lemma 18 we get that

$$\mathsf{Ex}\left[\sum_t 1_{E_t} \mid z\right] = \sum_{i=1}^{z} \frac{1}{i} \geq \int_1^{z+1} \frac{1}{x} = \ln(z+1),$$

and therefore

$$\mathsf{Ex}\left[\sum_t 1_{E_t}\right] = \mathsf{Ex}_z \mathsf{Ex}\left[\sum_t 1_{E_t} \mid z\right] \geq \mathsf{Ex}_z [\ln(z+1)].$$

Let $Z \sim \mathrm{Bin}(n, \alpha)$, and therefore we need to lower bound $\mathsf{Ex}[\ln(Z+1)]$ for $Z \sim \mathrm{Bin}(n, \alpha)$. To this end, we use the assumption that $\alpha > 1/n$, and therefore $\Pr[Z \geq \alpha \cdot n] \geq 1/4$ (see Greenberg and Mohri, 2013 for a proof of this basic fact). In particular, this implies that $\mathsf{Ex}[\ln(Z+1)] \geq \frac{1}{4} \ln(\alpha n + 1) > \frac{1}{4} \ln(\alpha n)$, which finishes the proof. □

Lemma 21 implies Theorem 5 as follows: set $k = d$, $k_1 = \cdots = k_d = 1$ and $n \geq 100k \log(1/\delta)$. Pick a distribution $D$ which is uniform over items, each satisfying exactly one of $d$ properties, and with value drawn uniformly from $[0, 1]$.

It suffices to show that with probability of at least 1/3, the algorithm retains an expected number of $\Omega(\log\log(1/\delta))$ items from a constant fraction, say 1/4, of the properties $i$. This follows from Lemma 21 as we argue next. Let $n_i$ denote the number of observed items of property $i$. Then, since $\mathsf{Ex}[n_i] = n/d = n/k \geq 100$, the multiplicative Chernoff bound implies that $n_i \geq n/2k \geq 2 \log(1/\delta)$

with high probability (probability = 1/2 suffices). Therefore, the expected number of properties $i$'s for which $n_i \geq 2\log(1/\delta)$ is at least $k/2$. Now, consider the random variable $Y$ which counts for how many properties $i$ we have $n_i \geq 2\log(1/\delta)$. Since $Y$ is at most $k$ and $\mathsf{Ex}[Y] \geq k/2$, then a simple averaging argument implies that with probability of at least 1/3 we have that $Y \geq k/4$. Conditioning on this event (which happens with probability $\geq 1/3$), Lemma 21 implies[5] that $\bar{\mathsf{Ex}}[|S_i|] = \Omega(\log\log(1/\delta))$ for each of these $i$'s.