[Reviews · NeurIPS 2019]

Reviewer 1



The paper is extremely well written, very nice read! Still, the connection to hiring is a bit tenuous. I understand motivating this as related to the secretary problem, but with types / multiple open positions, but the abstract/intro are really application-focused for what quickly becomes an entirely theory matching paper. I'm positive on the matching results, but perhaps the connection to real hiring is a bit oversold. Some more in-depth comments/questions follow in the Improvements section.

Reviewer 2



The paper presents very solid theoretical result. It is also related to a classical problem in online algorithm (the online secretary problem). Overall, the paper is nicely written. There is a couple of suggestions that I have, but nothing major. Line 43 and 44 could be clarified a bit. First, it is said that they consider "algorithms" (plural), but on line 44 is used only "algorithm" (singular). Also, in either case, what are those algorithms considered there? I would suggest rewriting that entire paragraph. "As we shall see in the next section, learning from the training set allows one to retain exponentially less items than is implied by the theorem above" -- I don't see how is this the case. Both of the results are linear in k. Was is the idea to say that there is an exponential improvement in 1/delta? The paper does not study a pure online question. In online settings, one is supposed to commit to a decision immediately, while here it is not the case that a candidate should be assigned/hired immediately. The studied setting in this paper falls more into streaming setting, where one makes a pass over the stream, collects a small number of elements and at the end of the stream performs final computation. Moreover, there has been a number of papers studied streaming with random arrivals (submodular maximization, computing matching and estimating their size, randomized greedy for maximal independent set ...). It would be nice to mention this in the paper. I find this paper as a good fit for the program of NeurIPS. As discussed already, it would be beneficial to comment about fallback options. --- Updated review --- Thank you for the feedback. The feedback addresses my "fallback" question, so I increased the score. It still would be nice to be more careful in which way is this problem addressed -- as already discussed, this problem does not fall into the standard category of online problems, so I hope that this will be addressed/discussed in a revised version.

Reviewer 3



my main concerns about the paper are: (1) Note that the authors considered d properties instead of one in previous research with the complex feasibility constraints. This makes the problem rather challenge and practical. However, the method the authors adopted for the d scenarios is relatively simple due to an independent assumption on the d properties, which may weaken the contribution here. (2) Is it meaningful to provide a bound on the minimum possible number of retained items? (See line 16) Is this a typo? In my opinion, it is more practical and meaningful to provider a bound on the maximum possible or expected number of retained items. (3) How to set the number of considate items, k_1, k_2, ..., k_d with regards to each properties in the first stages? And also, how to set the total condidate number k here? (4) In line 113, the notation for the edge (l, r) is legal, where l \in {1,2, ..., k}, r \in {1,2, ..., n}. However, the notation for v_l(r) represents the value of item r associated with property P_l, where l \in {1,2, ..., d}. There exists inconsistence. (5) In line 196-197, Assume towards contradiction that T retains an item cj \notin S..., should it be "reject" instead of "retain"? (6) The paragraph for "Formulation as bipartite matching" seems redundant here since it is no longer referred in the later part.

[Author Response · NeurIPS 2019]

We thank the reviewers for their thorough reviews, helpful comments, and for the various references and related areas they pointed us to. We will address all issues raised by the reviewers, add the missing references and clarify ambiguous paragraphs. Below we address the reviewers' main concerns.

**Reply to reviewer 1** We thank the reviewer for suggesting additional applications for motivating this problem. We will put more emphasis on the alternative motivations and discuss their suggested applications.

The reviewer writes: *"Interviews in practice are used to learn more about the candidate and judge their fit for the position. In this case $v_i(c)$ is therefore noisily learned during the interview process, not when they arrive..."*

In our description of the hiring example (second paragraph of the intro), the first screening phase where candidates are retained/discarded corresponds to processing the candidates files **before** deciding who to summon for an interview. Thus, the values $v_i(c)$ are determined by the candidates application (CV, skills, reference letters, etc.) and not by the interviews. We agree that during the interview more information about the candidate is gathered, and that this information is likely affect the values $v_i(c)$. Indeed, in a more realistic model for the hiring problem the values $v_i(c)$ should be updated after the interviews and before the final recruitment is computed. In other motivating examples (such as internet advertising or job scheduling) the values $v_i(c)$ can be assumed to be given. We believe that our abstraction does capture the essence of the problem at hand, and is reasonable for a variety of applications.

**Reply to reviewer 2** The reviewer writes: *"This question somewhat follows a new research topic called augmented-learning algorithms. Only in augmented-learning algorithms, there is usually also a fallback option?. I would be interested in seeing how they design an algorithm that performs quite well when the training set matches the actual hiring process, but does not perform arbitrarily bad when the training set is completely off of the actual candidates."*

Thanks for pointing out the connection and for suggesting the question. Indeed, our algorithm may fail when the thresholds computed in the learning phase are too large such that no item from the real-time data passes them. One simple fix is the following: after processing, say $n/2$, items from the real-time data, check whether the number of items retained by the threshold policy is too small than expected, and if this is the case then discard the threshold policy and apply the greedy algorithm from Theorem 1 on the remaining $n/2$ items. One can show that modifying the algorithm along these lines yields a "fallback" to the greedy oblivious setting from Theorem 1 in the (unlikely) case when the learned threshold policy is bad.

The reviewer writes: *" ... I don't see how is this the case. Both of the results are linear in $k$. Was is the idea to say that there is an exponential improvement in $1/\delta$?"*

We meant to refer in that sentence to the dependence on $n$ - the total number of items: the bound in Theorem 1 is proportional to $\log(n)$ (which is tight, by Theorem 2), while in Theorem 4 this bound is proportional to $\log\log(n)$. We will clarify this sentence.

**Reply to reviewer 3** Major concerns:

(1) *"...the authors considered $d$ properties instead of one in previous research with the complex feasibility constraints. This makes the problem rather challenge and practical. However, the method the authors adopted for the $d$ scenarios is relatively simple due to an independent assumption on the $d$ properties, which may weaken the contribution here."*

We remark that we do not assume anything about the distribution of the d properties per item, and the joint distribution on the properties and value per item is arbitrary. (We do assume that the marginal distribution over the values is continuous, i.e., atomless). Additionally, note that our algorithm treats the $d$ properties independently by design and that our method retains a near-optimal number of candidates (due to Theorem 5). We consider the simplicity of our solution as a strength rather than a weak contribution.

(3) *"How to set the number of candidate items, $k_1, k_2, ..., k_d$ with regards to each property in the first stages? And also, how to set the total candidate number k here?"*

The numbers of $k_1, \ldots, k_d$ and $k$ are given as part of the problem formulation; they are not set up by the algorithm but rather are given as an input.

Minor concerns:

(2) We refer to the number of items retained by the **best** algorithm; i.e., the one that retains as few items as possible while satisfying the desired optimality guarantee. We will try to clarify.

(5) No, recall that our goal is to prove that $T$ retains exactly the items in $S$ and so we want to rule out the possibility that it retains an item outside $S$.

(6) The main purpose of this formulation is demonstrate the connection with matching problems.

[Meta-Review · NeurIPS 2019]

The reviewers are in agreement that they liked the paper; but felt that the main section of the paper, while interesting, did not properly follow the abstract and introduction; and also suggested discussion of some additional references.